# A Review of Biomaterials Based on High-Entropy Alloys

Thiago Gonçalves de Oliveira [1], Danilo Valim Fagundes [1], Patrícia Capellato [2,*], Daniela Sachs [2] and Antonio Augusto Araújo Pinto da Silva [1]

[1] Institute of Mechanical Engineering, UNIFEI-Federal University of Itajubá, Av. BPS 1303, Itajubá 37500-903, Brazil

[2] Centre for Studies, Research and Innovation in Bifunctional Materials and Biotechnology, Institute of Physics and Chemistry, UNIFEI-Federal University of Itajubá, Av. BPS 1303, Itajubá 37500-903, Brazil

\* Correspondence: pat_capellato@yahoo.com.br

**Abstract:** Due to its great amount of microstructure and property possibilities as well as its high thermodynamic stability and superior mechanical performance, the new class of material known as high-entropy alloys (HEAs) has aroused great interest in the research community over the last two decades. Recent works have investigated the potential for applying this material in several strategical conditions such as high temperature structural devices, hydrogen storage, and biological environments. Concerning the biomedical field, several papers have been recently published with the aim of overcoming the limitations of conventional alloys, such as corrosion, fracture, incompatibility with bone tissue, and bacterial infection. Due to the low number of available literature reviews, the aim of the present work is to consolidate the information related to high-entropy alloys developed for biomedical applications (bioHEAs), mainly focused on their microstructure, mechanical performance, and biocompatibility. Topics such as phases, microstructure, constituent elements, and their effect on microstructure and biocompatibility, hardness, elastic modulus, polarization resistance, and corrosion potential are presented and discussed. The works indicate that HEAs have high potential to act as candidates for complementing the materials available for biomedical applications.

**Keywords:** high-entropy alloys; biomaterials; biocompatibility




## 1. Introduction

In the early 2000's, a new class of material was developed and aroused interest in the research community [1,2], leading to an exponential number of publications in the last two decades. They are widely known as high-entropy alloys (HEAs) or multi-principal element alloys (MPEAs). Although the precise definition of an HEA is still controversial, it seems to be consensus among authors that these alloys should be composed of at least four main elements with concentrations between 5 and 35 at% [2–7], in contrast to conventional alloys that are based on a main element (e.g., Fe for steel, Ni or Co for superalloys, Cu for bronze and brass, etc.). Considering the different possibilities of allowing elements and compositions, this class of material enables a large number of microstructures, applications, and properties [3,8]. HEAs can also be defined by their configurational entropy, which can be explained as a thermodynamic concept that defines the disorder of a system, according to Equation (1):

$$\Delta S_{config} = k * ln(w) \tag{1}$$

where $\Delta S_{config}$ is the configurational entropy, *k* is Boltzmann's constant ($1.38 \times 10^{-23}$ J/K), and *w* is the number of ways that the available energy can be mixed or shared between the particles in the system.

In this definition, increasing the number of elements leads to an increase in configurational entropy. For high-entropy alloys, the value of this property is at least 1.61R [8,9]. The thermodynamic properties of high-entropy alloys, their derived properties and the four

core effects of HEAs (high-entropy effect, sluggish diffusion, severe lattice distortion, and cocktail effect) are beyond the scope of the present article and are very well described in the textbook by Murty et al. [8].

In 2019, the same group published two articles [10,11] using the term bioHEA for the first time, as far as the authors know. Since then, this term has been applied for multi-principal element alloys that have been considered for applications in the biomedical field. The challenge is to overcome the limitations of conventional alloys (cp-Ti, Ti6Al4V, 316L, and CoCrMo alloys), and some promising results were reported concerning superior or similar corrosion resistance and implant degradation in a physiological environment [12–14], mechanical performance combined with biocompatibility [14–20], ion release [21,22], magnetic susceptibility [23,24], wear resistance [12,25], and bacterial infection [19].

In terms of mechanical properties, this review focuses on hardness and Young's modulus, which are considered critical for these application. Other mechanical properties are not discussed with the same importance in the publications presented in this review. Hardness is one of the most relevant properties for comparing materials, as it is easy to obtain [26]. In applications involving bone tissue, the hardness of the material must be equal to or greater than bone; otherwise, it will result in bone penetration. Furthermore, hardness is important for reducing the incidence of wear [27]. The Young's modulus is related to the stiffness of the material. Using the application example above, its value must be close to the value of the bone to avoid the stress shield effect and to avoid fracture and failure of the biomaterial [27,28]. Table 1 presents some mechanical properties of bone and conventional alloys.

**Table 1.** Mechanical properties of cortical bone and conventional alloys.

| | Young's Modulus (GPa) [Ref.] | Hardness (HV) [Ref.] | Yield Strength (MPa) [Ref.] | Tensile Strength (MPa) [Ref.] |
|---|---|---|---|---|
| Cortical bone | 10–30 [17,29] | - | 100–200 [29] | - |
| cp-Ti | 90–110 [17] | 120–200 [30] | 170–310 [1] [31] | >240 [31] |
| Ti6Al4V | 100–110 [28] | 310 [30] | 850–900 [32] | 860 [32] |
| 316L | 200 [29] | 130-160 [33] | 200–700 [29] | 480–1000 [30] |
| CoCrMo alloy | 240 [29] | 298 [33] | 450–1500 [29] | 655–1192 [32] |

[1] Grade 1.

As previously stated, several elements can be used for the composition of HEAs in order to obtain the desired properties, demonstrating good compositional freedom. Table 2 presents the most recurrent elements in bioHEAs compositions, based on the articles assessed of the present review. In small amounts and consequently not present in the table, the others elements presented in the articles include: Ni, Al, Mn, Si, B, Cu, V, Zn, W, Ga, Sn, Ag, Ca, Mg, Sr, Pd, and Yb. It should be noted that some elements such as nickel have carcinogenic potential, according to the International Agency for Research on Cancer (IARC) [34].

**Table 2.** Recurrence of elements in bioHEAs compositions evaluating 49 selected publications.

| Elements | Ti | Zr | Nb | Ta | Mo | Hf | Fe | Cr | Co |
|---|---|---|---|---|---|---|---|---|---|
| Recurrence | 57 | 49 | 48 | 44 | 28 | 27 | 12 | 12 | 12 |

Regarding the development of bioHEAs, most of the experimental works have used melting techniques (Figure 1), as they are more suitable techniques for the fusion of reactive elements such as Ti, Zr, and Hf. Some papers have indicated that the induction technique leads the microstructure to a structure similar to that of arc melting, as is the case in a publication by Nagase et al. [11]. Techniques using powder, such as powder metallurgy and selective laser melting (SLM), are prominent routes for enabling greater structural

homogeneity. In addition, the application of computational thermodynamics and ab initio calculations to predict the crystal structure and properties of HEAs is highlighted, due to the difficulty in predicting these characteristics for these alloys due to the large number of elements with significant amounts.

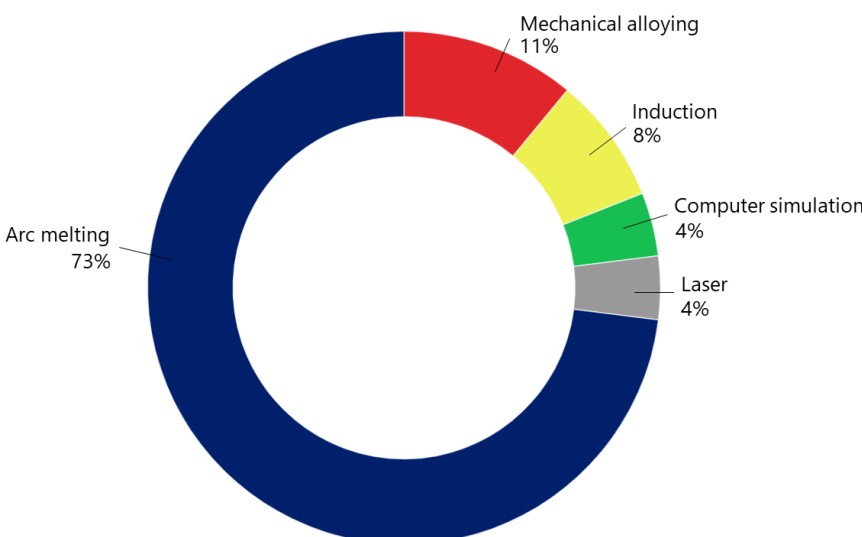

**Figure 1.** Percentage of bioHEAs development techniques evaluating 49 selected publications.

Two very interesting reviews in the literature were found involving high-entropy alloys for biomedical applications. Castro et al. [35] is a review with a main focus on surveying potential applications, mechanical performance, and has a brief section on biocompatibility studies. Ahmady et al. [36] presents a review focused on bioHEA coatings. The objective of this review is to complement the existing literature, with a focus on microstructural characteristics and biological and chemical properties.

## 2. Microstructure and Mechanical Performance

### 2.1. BioHEAs with Single-Phase BCC

The review on high-entropy alloys as biomaterials, with the intention of using them for the varied medical applications (implants, stents, structures, and coatings of conventional alloys), shows that the vast majority have a BCC structure. Generally, refractory elements with Ta, W, Nb, Mo, or V are BCC-phase stabilizers [18,19,37].

Based on the publications, Table 3 brings together the alloys, Vickers hardness, and Young's modulus of high-entropy alloys for biomedical purposes that have only one BCC phase. In order to make it possible to compare the different works, all hardness values in the GPa unit were converted to HV (Equation (2)).

$$HV = \frac{\text{Value in GPa}}{0.009807} \qquad (2)$$

**Table 3.** Mechanical properties of bioHEAs with single-phase BCC.

| Alloy | Route/Method | Post-Processing | Hardness (HV) | Young's Modulus (GPa) | Reference |
|---|---|---|---|---|---|
| $(Ti_{0.3}Zr_{0.3}Hf_{0.3})_{15}(Nb_{0.5}Ta_{0.5})_x$ [1] | Arc melting | HT [2] 1200 °C-24 h | 287–293 | 56–68 | [38] |
| TiNbZrHfTa | Arc melting | HT [2] 790 °C-1 h | - | 66 | [19] |
| TiTaHfNb | | | - | 112 | |
| TiTaHfNbZr | Arc melting | - | - | 132 | [37] |
| TiTaHfMoZr | | | - | 159 | |
| TiNbZrTaHf | Mechanical alloying | - | 564 | 79 | [39] |
| HfNbTaTiZr | Computational method | - | 297 | 97 | [40] |
| $Hf_{0.5}Nb_{0.5}Ta_{0.5}Ti_{1.5}Zr$ | | | 253 | 86 | |

**Table 3.** *Cont.*

| Alloy | Route/Method | Post-Processing | Hardness (HV) | Young's Modulus (GPa) | Reference |
|---|---|---|---|---|---|
| $(MoTa)_x NbTiZr$ [3] | VAM | HT [2] 1400 °C-4 h | 380–430 | 113–125 | [41] |
| $Ti_{1.4} Nb_{0.6} Ta_{0.6} Zr_{1.4} Mo_{0.6}$ | SLM | - | - | 140 | [15] |
| $TiZrHfCr_{0.2} Mo$ | Arc melting | - | 531 | - | [42] |
| $TiZrHfCo_{0.07} Cr_{0.07} Mo$ | | | 532 | - | |
| $HfNbTaTiZr$ | Arc melting | - | 320 | 112 | [12] |
| $Hf_{0.5} Nb_{0.5} Ta_{0.5} Ti_{1.5} Zr$ | | | 307 | 98 | |
| $Ta_x Nb_x HfZrTi$ [4] | Arc melting | - | - | 73–103 | [23] |
| $HfNbTaTiZr$ | Powder techniques | HPT [5] (2.5 GPa) | 410 [6] | - | [43] |
| $TiMo_{20} Zr_7 Ta_{15} Si_{0.5}$ | | | 337 | 89 | |
| $TiMo_{20} Zr_7 Ta_{15} Si_{0.75}$ | VAR | - | 355 | 69 | [44] |
| $TiMo_{20} Zr_7 Ta_{15} Si_{1.0}$ | | | 356 | 79 | |
| $TiZrHfNbTa$ | Arc melting | - | - | - | [45] |

[1] For x = 3 and 5. [2] Heat treatment. [3] For x = 0.2, 0.4 and 0.6. [4] For x = 0.4, 0.5, 0.6, 0.8 and 1. [5] High pressure torsion. [6] Value obtained after refinement by HPT.

Through high-pressure torsion (HPT) processing, the HEA TiNbZrTaHf of BCC structure [39] was subjected to severe plastic deformation for significant grain refinement. A high density of defects in microstructure was observed, with a crystallite size below 100 nm. Regarding mechanical performance, the alloy stands out among the other single-phase BCC bioHEAs, with a hardness of 564 HV. This value is considerably above the values presented by conventional alloys such as Ti6Al4V (340–345 HV) and 316L (228 HV) (Table 1). The elastic modulus of 79 GPa is also highlighted as presenting a lower value compared to Ti6Al4V (120 GPa) and cp-Ti (90–110 GPa) (Table 1). These characteristics may favor applications in implants, due to the need for a Young's modulus close to the value of the bone. The cocktail effect can partially describe the high hardness of the composition mainly due to the presence of Hf and Ta, but it does not explain the low Young's modulus. According to the authors, these characteristics may be related to the binding energy of the elements [39].

In the same way, one of the alloys studied by Málek et al. [43] (HfNbTaTiZr) was refined by HPT. The HEA, processed by the spark plasma sintering (SPS) method and refined, presented BCC structure with a dense and fragile sample. Microstructure refinement resulted in a 410 HV and grain size less than 500 nm. Consequently, HPT technique is suitable for the refinement of high-entropy alloys, with good results for microstructure and mechanical properties.

Segregation of elements in high-entropy alloys is a problem even for single-phase alloys. Based on this, Akmal et al. [41] used the remelting process for HEA $(MoTa)_x NbTiZr$ in order to homogenize the elements and eliminate the dendritic structure. For x = 0.2, 0.4, and 0.6, the BCC phase was obtained, while for 0.8 and the equiatomic proportion of the composition (x = 1), the presence of two BCC phases was verified. It was described that as Mo and Ta were added, the solid solution hardened and there was a reduction in grain size (from about 1 mm without Mo and Ta to 80 μm equiatomic alloy), in addition to an increase in the Young's modulus. The hardness values for single BCC phase were between 380 and 430 HV, while the elastic modulus values calculated by matrix are between 110 and 125 GPa, as can be seen in Figure 2a [41].

Following the same logic, the increment in the content of Ta and Nb led to an increase in the Young's modulus of the HEA $Ta_x Nb_x HfZrTi$ (x = 0.2, 0.4, 0.5, 0.8, and 1) and it decreased in only one case (x = 0.6) [23]. The values found for the elastic modulus ranged from 73 to 103 GPa (Figure 2b). Among the proportions, only the one containing x = 0.2 did not have a single BCC, resulting in BCC and HCP [23].

For HEA $TiMo_{20} Zr_7 Ta_{15} Si_x$ [44], it was found that the addition of Si content increased the hardness value. For x = 0.5, the hardness was 337 HV and the Young's modulus was 89 GPa. With x = 0.75, 355 HV and 69 GPa. Furthermore, for x = 1, the hardness reached a value of 356 HV, and the elastic modulus also increased to 79 GPa. Ti and Zr tended to enrich in the interdendritic region, while Ta and Mo were more abundant in the dendrites [44].

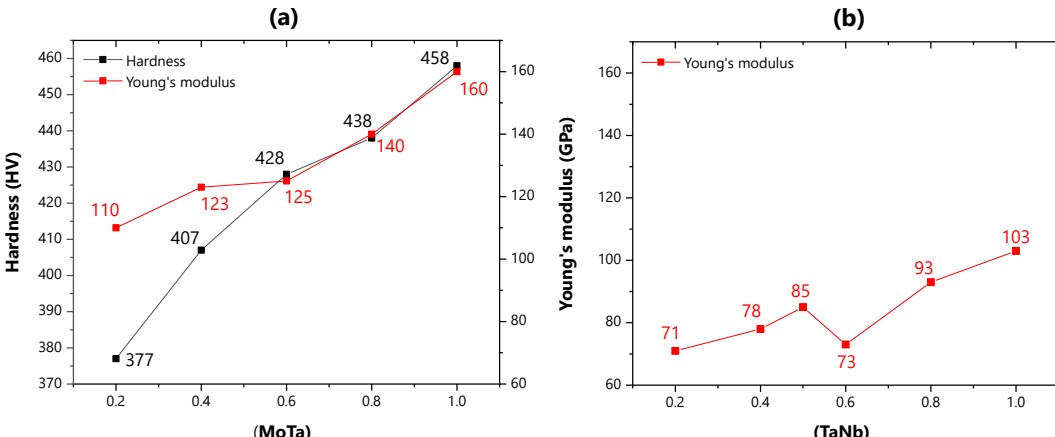

**Figure 2.** (**a**) Evolution of hardness and Young's modulus as a function of MoTa ratio for $(MoTa)_x NbTiZr$. Based on [41]; (**b**) Evolution of Young's modulus as a function of Ta and Nb ratio for $Ta_x Nb_x HfZrTi$. Based on [23].

The publications by Bhandari et al. [40] and Motallebzadeh et al. [12] allow an important comparison between computational methods and experimental results for the HfNbTaTiZr and $Hf_{0.5}Nb_{0.5}Ta_{0.5}Ti_{1.5}Zr$ alloys. The experimental work [12] confirmed the expected results from the theoretical calculations (density functional theory—DFT) [40] for the microstructure (both single-phase BCC). However, for mechanical properties, a certain difference was obtained between values of hardness and elastic modulus. For example, for equiatomic HEA, the predicted result was 297 HV and the Young's modulus was 97 GPa. The experimental work showed 320 HV and 112 GPa. For the second alloy $(Hf_{0.5}Nb_{0.5}Ta_{0.5}Ti_{1.5}Zr)$, the predicted result was 253 HV and 86 GPa, while 307 HV and 98 GPa were the experimentally indicated values.

Three TiTaHf-based HEAs (TiTaHfNb, TiTaHfNbZr, and TiTaHfMoZr) were studied by Gurel et al. [25,37,46]. The authors indicated that the addition of Mo and Zr resulted in a reduction in ductility when compared to the four-element alloy. Regarding the elastic modulus, the TiTaHfNb, TiTaHfNbZr, and TiTaHfMoZr alloys have a lower elastic modulus than materials commonly used for implants: 112 GPa, 132 GPa, and 159 GPa, respectively. Furthermore, the addition of Mo resulted in a greater heterogeneity of the microstructure [37].

Studying an HEA with the same composition (TiNbZrHfTa) of one of the alloys studied by Gurel et al. [25], the works of Yang et al. [45] and Berger et al. [19] confirmed the single-phase BCC microstructure. However, the elastic modulus value was much lower than that found by Gurel et al. [25]: 66 GPa [19]. For this alloy, a tensile strength of 1050 MPa was obtained. This value is close to the property values for 316L and CoCrMo alloys and higher than the that of Ti6Al4V, as shown in Table 1.

It is worth noting that the SLM technique is important in order not to obtain double BCC. Manufactured by SLM, the HEA sample $Ti_{1.4}Nb_{0.6}Ta_{0.6}Zr_{1.4}Mo_{0.6}$ [15] showed a porosity of less than 0.5%. It is explained that during the solidification of the bulk, the cooling rate is the most important factor. Thus, it is assumed that a high cooling rate of SLM prevents extensive elemental segregation. The sample presented a BCC structure, with a dendritic phase rich in Nb, Ta, and Mo, and an interdendritic phase rich in Ti and Zr. The Young's modulus of the alloy is 140 GPa [15].

## 2.2. BioHEAs with Dual Phase BCC

Table 4 shows the mechanical properties of high-entropy alloys for biomedical applications that have double BCC, with the composition MoNbTaTiZr being a priority.

**Table 4.** Mechanical properties of dual BCC bioHEAs.

| Alloy | Route/Method | Hardness (HV) | Young's Modulus (GPa) | Reference |
|---|---|---|---|---|
| $(MoTa)_{0.8}NbTiZr$ | VAM | 480 | 140 | [41] |
| MoTaNbTiZr | | 510 | 160 | |
| $Ti_x ZrNbTaMo$ [1] | Arc melting | 430–490 | - | [20] |
| TiZrNbTaMo | Induction | 619 | - | [47] |
| $Ti_{30}(NbTaZr)_{60}Mo_{10}$ | | 487 | - | |
| MoNbTaTiZr | VAM | 657 | 164 | [13] |
| TiNbTaMoZr | Mechanical alloying | 591 | 62 | [48] |
| TiZrNbTaMo | Computational method | - | 122–144 | [49] |
| TiZrNbTaMo | Arc melting | 500 | 153 | [50] |

[1] For x = 0.5, 1, 1.5 and 2.

Although used in smaller numbers for bioHEAs, powder metallurgy is an important technique for biomedical applications due to its known ability to reduce Young's modulus. Another important factor is the fact that they do not present a severe segregation effect, unlike the HEAs casting that will be discussed below. We analyzed two works involving the equimolar alloy MoNbTaTiZr. In the first [48], the alloys were milled in a time between 1 and 20 h, selecting a time of 10 h as the optimal milling time. Subsequently, the alloys were treated for 1 h at temperatures between 1450 and 1500 °C. Figure 3a shows the SEM-EBSD image treated at 1450 °C for 1 h. In the second work [18], powder was prepared by hydrogenation–dehydrogenation followed by plasma spheroidization and SPS at 1400 °C and 50 MPa for 15 min. Figure 3b shows that the application of the SPS method provided a coarser microstructure. Regarding the properties, Akmal et al. [18] did not report the values obtained in the compression tests and did not indicate hardness values, however it is possible to observe a yield point between 1500 and 2000 MPa. This is a significant value even when compared with the CoCrMo alloys (Table 1). Finally, the work by Normand et al. [48] showed a high hardness of 591 HV and a Young's modulus of 62 GPa. The elastic value of HEA is well below the values known for conventional biomedical alloys, which suggests better performance in bone implants, for example.

Another four MoNbTaTiZr equimolar HEAs were studied in different works, three of them manufactured by arc melting [13,20,50] and one by induction [47]. For the works by Hua et al. [20] and Wang and Xu [50], a hardness of approximately 500 HV was obtained, while for the work of Shittu et al. [13] and Li et al. [47], higher values were found: 657 and 619 HV, respectively. The reason for this divergence is not clear, but factors such as different grain sizes resulting from different cooling rates and the degree of segregation may be determinant for such different values. Wang and Xu [50], for example, indicates segregation of Ta, Mo, and Nb in the dendritic arms. Hua et al. [20] did not disclose the elastic modulus values; however, Shittu et al. [13] and Wang and Xu [50] reported relatively close values.

With the variation in the Ti content for the double BCC HEA TiZrNbTaMo [20], it was possible to carry out a study on the microstructure (Figure 4) and the mechanical performance of the alloy. The proportions of Ti were 0.5, 1, 1.5, and 2. During solidification, the main dendritic phase presented elements with high melting temperature (Ta and Mo), while Ti and Zr were segregated from this phase and enriched in the interdendritic region. Therefore, this region was formed by elements with low melting points. The explanation of the distribution of the elements by the heat of mixing is suggested: between Ta, Nb, Ti, and Zr there is the positive heat of mixing, and Ti and Zr are first separated from the dendritic phase and enriched in the interdendritic region. Finally, it was observed that an decrease in Ti content coincides with an increase in hardness, with values from 430 to 490 HV. On the other hand, increasing Ti content induces a lower yield strength: 1580 MPa for the alloy with $Ti_{0.5}$ and 1440 for $Ti_2 ZrNbTaMo$ (Figure 5). The HEA $Ti_{0.5} ZrNbTaMo$ stood out for its hardness, high compressive strength (2600 MPa), and plastic deformation of more than 30%. The authors do not disclose how many measurements were taken from the microhardness

tests. On the other hand, the Young's modulus and the yield strength were determined by compression tests in just one sample [20].

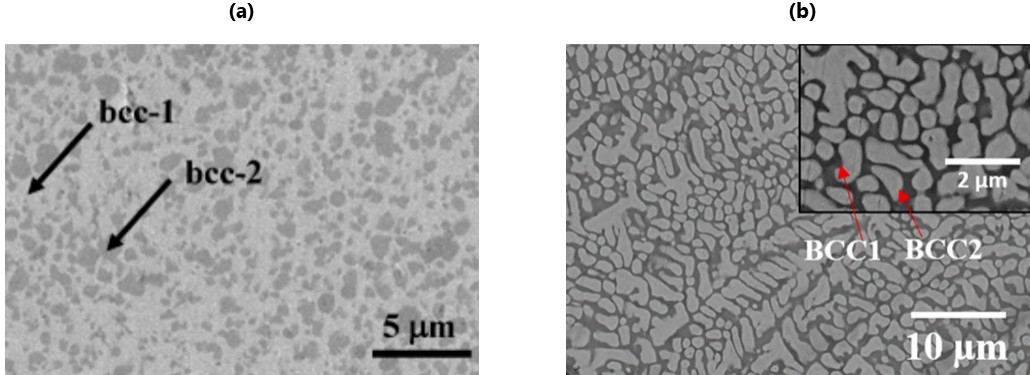

**Figure 3.** (**a**) SEM image showing the two BCC regions for HEA TiNbTaMoZr (from [48]); (**b**) SEM image of HEA MoNbTaTiZr showing the BCC1 region (dark region) and BCC2 (light region) (from [18]). Reprinted with permission from Elsevier: Mater. Chem. Phys. Copyright 2022, License: 5350300554948.

As highlighted in the previous section, the addition of Mo and Ta leads to an increase in hardness and elastic modulus. From the content of 0.8 for these elements, the double formation of BCC was obtained for bioHEA $(MoTa)_x NbTiZr$ [41], with hardness values of 480–510 HV (x = 0.8 and 1). Analyzing the $Ti_{30}(NbTaZr)_{60}Mo_{10}$ [47], we observed that the hardness is 487 HV, close to the values found by Akmal et al. [41], although the proportion of each element is different.

Koval et al. [49] used a computational method in order to study the TiZrNbTaMo alloy to be applied in the biomedical area [49]. They applied the DFT for elastic properties and the USPEX method to predict the crystal structure, obtaining a BCC structure. In their results, they argue that the chemical composition and type of lattice are more important for the elastic properties than the arrangement of lattice atoms. The calculated elastic modulus values were between 122 and 140 GPa, from the variation in the total number of atoms in the cell.

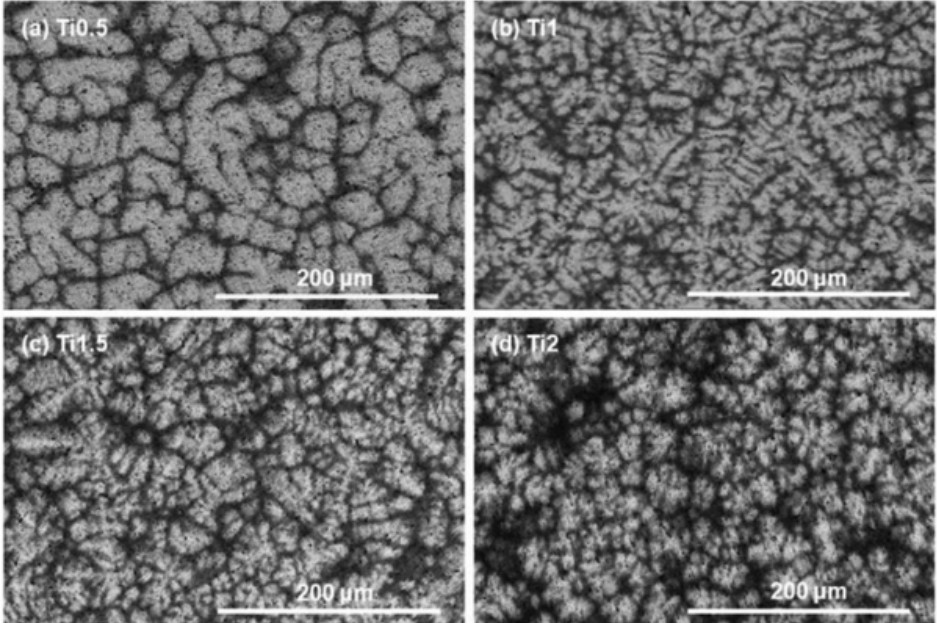

**Figure 4.** SEM images of alloys (**a**) $Ti_{0.5}ZrNbTaMo$, (**b**) TiZrNbTaMo, (**c**) $Ti_{1.5}ZrNbTaMo$ and (**d**) $Ti_2ZrNbTaMo$ (from [20]). Reprinted with permission from Elsevier: J. Alloys Compd. Copyright 2022, License: 5355940671199.

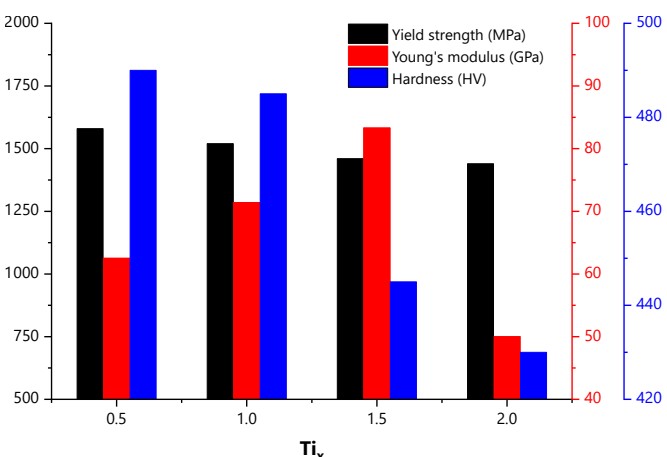

**Figure 5.** Evolution of yield strength, Young's modulus, and hardness for the Ti$_x$ZrNbTaMo alloy. Based on [20].

Several alloys cast with composition MoNbTaTiZr showed segregation of Ta, Nb, and Mo in the dendritic region, while Ti and Zr were mostly concentrated in the interdendritic region [50–53], as well as the HEA produced by induction [47]. What was commented on in the Introduction regarding the similarity between the microstructure of alloys produced by melting and induction is highlighted.

In the study by Perumal et al. [22], the stationary friction processing (SFP) method was used to homogenize the microstructure of the HEA MoNbTaTiZr. Fifteen minutes of processing in this manner allowed for significant homogenization of the elements in the dendritic and interdendritic regions.

A double BCC was obtained in two works with small differences from the compositions mentioned above. The first includes V (TiMoVNbZr) [54] and the second has the addition of W (TiNbTaZrW) [11].

### 2.3. BioHEAs with Single-Phase FCC or Dual FCC

Table 5 presents the routes and phases of high-entropy alloys with single-phase FCC and double FCC. It is noteworthy that only one of the works reported values of hardness and elastic modulus [16], which will be discussed below.

**Table 5.** BioHEAs with single-phase FCC or dual FCC.

| Alloy | Route/Method | Phases | Reference |
| --- | --- | --- | --- |
| CoCrFeCuNi | SLM | FCC | [9] |
| FeCoNiCrPd | VAR | FCC | [55] |
| Al$_{0.1}$CoCrFeNi | Arc melting | FCC | [16] |
| Al$_{0.4}$CoCrCuFeNi | Induction | Double FCC | [56] |
| AgCoCrFeMnNi | | | |
| CuCoCrFeMnNi | | | |
| CoCrCu$_2$FeMnNi | | | |
| CoCrCu$_3$FeMnNi | Arc melting | Double FCC | [57] |
| CoCrCuFeMnNiB$_{0.2}$ | | | |
| CoCrCu$_2$FeMnNiB$_{0.2}$ | | | |
| CoCrCu$_3$FeMnNiB$_{0.2}$ | | | |

For FCC and double FCC structure, a large presence of CoCrCu and FeNi was found in the formation of these phases in HEAs for biomedical applications.

The HEA CoCrFeCuNi [9], developed by SLM, has a single-phase structure (FCC) and uniform distribution of composition [9]. With the compression test, the average yield strength obtained was 516 MPa. This value can be compared to the range of property values for 316L and CoCrMo alloys. In an alloy of similar composition produced by VAR,

with the replacement of Cu by Pd [55], a single-phase FCC microstructure was obtained, but no mechanical properties were shown [55].

The FCC single-phase HEA $Al_{0.1}CoCrFeNi$ [16] underwent heat treatment of annealing (1000 °C for 24 h) and cold rolling, which improved the hardness of the material, reaching 143 HV. It was possible to obtain a better tensile strength (570 MPa) and yield strength (212 MPa). The tensile strength value is close to that of cortical bone (100–200 MPa), as shown in Table 1. The yield strength result shows lower values compared to Ti6Al4V, 316L, and CoCrMo alloys. However, the Young's modulus did not change with the thermomechanical process (203 GPa). A similar alloy, with the addition of Cu and having a higher content of Al ($Al_{0.4}CoCrCuFeNi$) [56], was developed in order to obtain a good combination of mechanical and antimicrobial properties. Its microstructure showed two FCC phases, as shown in Figure 6. The dendritic regions were enriched with Co, Cr, Fe, and Ni, while the interdendritic regions have mainly Cu. Mechanical tests were not performed for this bioHEA.

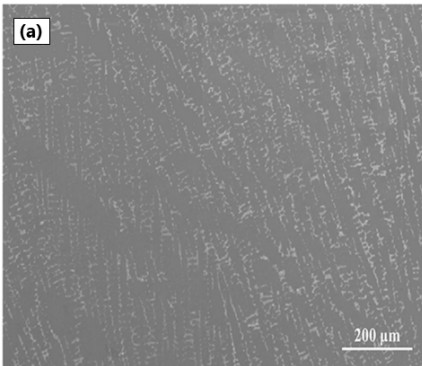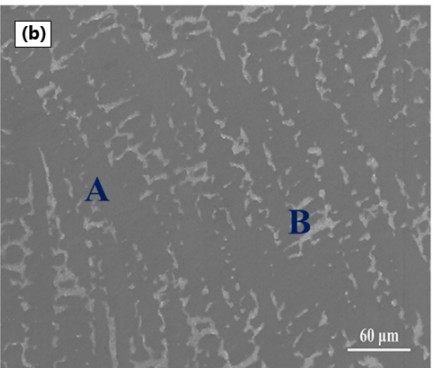

**Figure 6.** SEM images showing the microstructure of HEA $Al_{0.4}CoCrCuFeNi$ (**a**) melted and (**b**) after processing by HPT with dendritic and intedendritic regions (from [56]). Reprinted with permission from Elsevier: J. Mater Sci. Technol. Copyright 2022, License: 5355941102509.

Seven HEAs (AgCoCrFeMnNi, CuCoCrFeMnNi, $CoCrCu_2FeMnNi$, $CoCrCu_3FeMnNi$, $CoCrCuFeMnNiB_{0.2}$, $CoCrCu_2FeMnNiB_{0.2}$, and $CoCrCu_3FeMnNiB_{0.2}$) were evaluated for liquid phase separation (LPS) through heat of mixing and studies by thermodynamic calculations [57]. All HEAs exhibited double FCC and no phase-separated structure formed by LPS was observed in $CoCrCu_xFeMnNi$ (x = 1, 2, and 3). However, the addition of B increased the trend in liquid phase separation [57].

### 2.4. BioHEAs with Amorphous Phase

The amorphous structure obtained by sputtering can be attributed to the low enthalpy, high mixing entropy, slow diffusion, and the difference in atomic radii of the elements [58]. The hardness of HEAs with amorphous phase presents significantly higher values when compared to conventional BCC and FCC alloys, as expected and which can be seen in Table 6.

Of the HEAs that presented only an amorphous phase, there is a predominance of TiTaHf elements. It should be noted that practically all the works whose alloys showed amorphous microstructure involved melting and deposition onto substrates. This deposition technique has become quite popular due to the formation of a uniform layer of the coating with good adhesion and easy control of the composition and structure of the film [58].

Two studies evaluated the coating of Ti6Al4V alloys with HEAs using the sputtering technique. In the first, the $Ti_{1.5}ZrTa_{0.5}Nb_{0.5}W_{0.5}$ [58] coating obtained a hardness of 1835 HV and an elastic modulus of 210 GPa. Some samples received an incorporation of Ag nanoparticles, reaching a hardness of 1631 HV and 200 GPa in their elastic modulus values. For the alloy TiTaHfNbZr [14], the tribological properties were analyzed. A surface

that was protective against wear and cracks was found, which is relevant for implants in long-term and load-bearing applications, and can be exemplified in hip or knee joints. With the nanoindentation test, it was possible to obtain a hardness of 1276 HV and a Young's modulus of 181 GPa.

**Table 6.** Mechanical properties of amorphous phase bioHEAs.

| Film | Route | Substrate | Hardness (HV) | Young's Modulus (GPa) | Reference |
|---|---|---|---|---|---|
| Ti$_{1.5}$ZrTa$_{0.5}$Nb$_{0.5}$Hf$_{0.5}$ | Arc melting | 316L | 1165 | 180 | [59] |
| | | CoCrMo | 1172 | 185 | |
| | | Ti6Al4V | 1168 | 183 | |
| TiTaHfNbZr | VAM | Ti6Al4V | 1276 | 181 | [14] |
| TiTaHfNbZr | Arc melting | NiTi | 1285 [1] | 183 | [60] |
| | | NiTi | 1132 [2] | 173 | |
| TiTaHfNbZr | VAM | NiTi | 1285 | 183 | [21] |
| Ti$_{1.5}$ZrTa$_{0.5}$Nb$_{0.5}$W$_{0.5}$ HEA-9Ag NPs | Arc melting | Ti6Al4V | 1835 | 210 | [58] |
| | | Ti6Al4V | 1631 | 200 | |

[1] Value obtained for 750 nm thick film. [2] Value obtained for 1500 nm thick film.

Two other papers studied the deposition of TiTaHfNbZr on NiTi substrates. Motallebzadeh et al. [60] concluded that the grain size and surface roughness increased along with the thickness of the deposited film. With a smaller thickness (750 nm), a higher hardness was obtained: 1285 HV. The Young's modulus was 183 GPa. For the thickest film (1500 nm), a hardness of 1132 HV and an elastic modulus of 173 GPa were calculated. Aksoy et al. [21] showed values for hardness and the elastic modulus equal to those obtained for thinner films in the work by Motallebzadeh et al. [60]. The authors also comment that the similarity between elastic modules and microstructures results in strong adhesion between the substrate and the coating at the interface, allowing the application of stresses to be evenly distributed within the substrate and the coating, minimizing or completely eliminating the risk of delamination [21].

Three alloys commonly used for medical devices (316L, CoCrMo, and Ti6Al4V) were coated with HEA Ti$_{1.5}$ZrTa$_{0.5}$Nb$_{0.5}$Hf$_{0.5}$ [59], in order to evaluate the hardness of the film with the substrate and the microstructure of the coating using nanoindentation. All three alloys increased their hardness with the HEA coating. For 316L steel, the hardness increased from 248 to 1165 HV. CoCrMo presented the highest value: from 419 to 1172 HV. Finally, the 338 HV Ti6Al4V alloy reached 1168 HV.

### 2.5. Other Phases Obtained with bioHEAs

Finally, several publications point to others constituent phases, such as BCC and FCC, BCC and amorphous phase, BCC and HC, and primitive cubic phase (cP), among others that will be discussed below. Table 7 presents a summary of the works found with microstructures containing these combination of phases.

A non-equiatomic FeCoNiTiAl coating was applied to substrates of the porous Ti6Al4V alloy [61]. The spraying time was varied between 0.5, 1, 2, and 3 h. With this, it was possible to evaluate the changes in the mechanical properties and the quality of the coating (Figure 7). For the spraying times used, the Young's modulus exhibited values of 100 GPa (0.5 h), 132 GPa (1 h), 120 GPa (2 h), and 112 GPa (3 h). Regarding the hardness of the samples, values of 867 HV (0.5 h), 2294 HV (1 h), 1754 HV (2 h), and 1479 HV (3 h) were obtained. The samples with deposition times of 1 and 2 h showed higher Young's modulus and hardness values when compared to the others. This can be explained by the fact that increasing the sputtering time causes the spatter particles accumulated in the coating to increase, decreasing the quality of the coating. Therefore, when the spraying time was 1 h, the HEA coating showed superior quality and better mechanical properties. All samples have BCC and FCC structure [61].

**Table 7.** Mechanical properties of HEAs for biomedical applications.

| Alloy | Route | Structure/Phase | Hardness (HV) | Young's Modulus (GPa) | Reference |
|---|---|---|---|---|---|
| FeCoNiTiAl (0.5 h [1]) | | | 867 | 100 | |
| 1 h [1] | - | BCC and FCC | 2294 | 132 | [61] |
| 2 h [1] | | | 1754 | 120 | |
| 3 h [1] | | | 1479 | 112 | |
| $Al_{0.6}CoCrFeNi$ | | | 245 | - | |
| $Al_{0.8}CoCrFeNi$ | VAR | BCC and FCC | 427 | - | [62] |
| AlCoCrFeNi | | | 562 | - | |
| AlCrFeCoNi | | cP | 562 | - | |
| $AlCoCrFeNi_{1.4}$ | VAR | FCC and cP | 455 | - | [63] |
| $AlCoCrFeNi_{1.8}$ | | FCC and cP | 316 | - | |
| $(TiZrNb)_{14}SnMo$ | VAM | BCC and HCP | 551 | 110 | [64] |
| HEA coating | | | 584 | 89 | |
| TiAlFeCoNi | Arc melting | BCC and L2$_1$ | 635 | 250 | [65] |
| HEA-HPT | | | 880 | 126 | |
| $Ta_{0.2}Nb_{0.2}HfZrTi$ | Arc melting | BCC and HCP | - | 71 | [23] |
| TiNbMoMnFe | Powder metallurgy | BCC and amorphous | - | - | [66] |
| AgAlNbTiZn | Powder metallurgy | BCC and FCC | - | - | [67] |

[1] Sputtering time.

In the analysis of HEA TiAlFeCoNi [65], a BCC phase and ordered L2$_1$ were obtained. The alloy obtained by melting was subjected to high-pressure torsion (HPT) to improve hardness and grain refinement (Figure 8), increasing from 635 HV to 880 HV. As for the Young's modulus, it decreased from around 250 to 126 GPa [65].

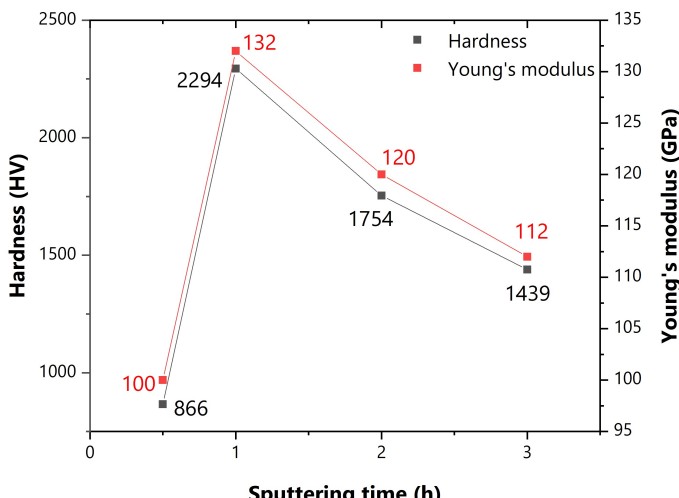

**Figure 7.** Evolution of hardness and Young's modulus as a function of sputtering time of the FeCoNiTiAl coating. Based on [61].

The non-toxic alloy $(TiZrNb)_{14}SnMo$ after VAM showed BCC structure and HCP dendrites [64]. In this work, laser coating was performed on pure Ti substrate. Due to the rapid solidification of this process, the authors reported a decrease in dendritic segregation and suppression of the HCP phase. The hardness of the HEA coating compared to the alloy itself is higher due to grain refinement and the supersaturated solid solution, reaching 584 HV and an elastic modulus of 89 GPa. On the other hand, the HEA obtained by VAM showed a hardness of 551 HV and 110 GPa in the raw solidification state [64].

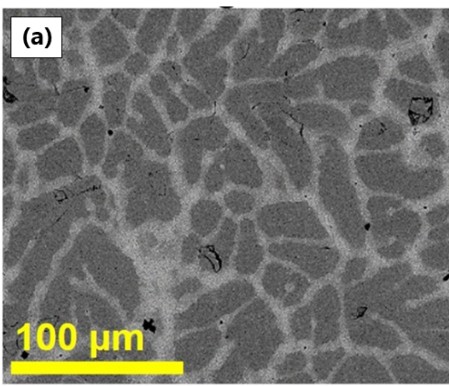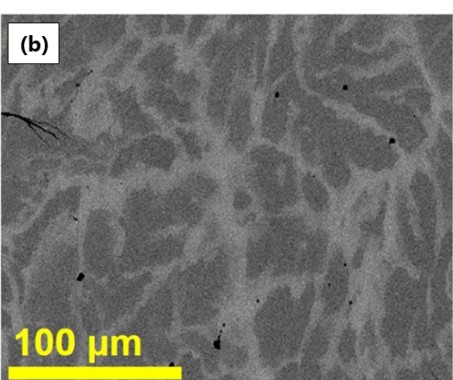

**Figure 8.** SEM images showing the microstructure of HEA TiAlFeCoNi (**a**) melted and (**b**) after processing by HPT (from [65]). Reprinted with permission from Elsevier: Mater. Sci. Eng., C. Copyright 2022, License: 5355941408547.

Two independent studies evaluated the influence of Ni [63] and Al [62] levels on AlCoCrFeNi-based bioHEAs produced by VAR. Rios et al. [63] indicated that the addition of Ni reduced the hardness of the alloy (from 562 to 316 HV, according to Table 7), due to the dissolution of the precipitates in a matrix rich in Ni and the formation of a solid solution. The extent of the interdendritic regions increased with higher proportions of Ni. Regarding the microstructure, the authors reported that the equiatomic alloy presented an unconventional primitive cubic phase (cP), while the others presented a fractional second-phase FCC [63]. In the work by Socorro et al. [62], for the same equiatomic alloy, the presence of the BCC phase was reported, while lower Al contents indicated stabilization of the FCC phase. On the other hand, the hardness values for the equiatomic alloy are compatible in the two works [62].

The influences of Ta and Nb stabilize the BCC phase, as can be seen in the $Ta_xNb_xHfZrTi$ [23] alloy. For a content of 0.2 for these elements, HEA presented BCC and HCP phases, with low elastic modulus (71 GPa) and yield strength of 480 MPa. With Ta and Nb contents between 0.4 and 1 (0.4, 0.6, 0.8, and 1) only BCC phase was obtained [23].

## 3. Biological and Chemical Properties

### 3.1. Anticorrosive Performance

Among the assays and tests that evaluated the biocompatibility for bioHEAs, anticorrosive performance is the most studied. The importance of evaluating this parameter is because it is a key factor for biocompatibility, in which corrosion resistance directly affects the functionality and durability of an implant material [68].

Table 8 presents the corrosion potential ($E_{corr}$), the polarization resistance ($R_p$) and the corrosion current density ($I_{corr}$) for comparison between conventional biomedical alloys and high-entropy alloys. It should be noted that each group of authors used different parameters for the corrosive tests, which makes it difficult to directly compare the results obtained for high-entropy alloys. Phosphate buffer solution (PBS), fetal bovine serum (FBS), Ringer's solution, simulated body fluid (SBF), NaCl solution, and Hank's solution were used as solutions that the samples were to be exposed to. Furthermore, as explained by Eliaz [68], temperature and pH are factors that influence the corrosion behavior of materials, which may also explain the difference in values for alloys with the same compositions and that were used in the same solution. Two examples of this case include the work of Navi et al. [52] and Shittu et al. [13].

Several studies have compared HEAs with conventional alloys Ti6Al4V, CoCrMo, and 316L. Regarding the corrosion potential, Motallebzadeh et al. [12] evaluated two TiZrTaHfNb-based alloys with different compositions. These alloys presented a lower performance compared to Ti6Al4V, but with better polarization resistance. The authors argue that it may have been due to the higher content of electronegative elements, such as

Ti and Zr. In contrast, the work by Yang et al. [45] showed a lower corrosion potential for HEA TiZrTaHfNb in relation to Ti6Al4V and lower resistance to polarization.

Wang and Xu [50] and Navi et al. [52] studied a similar HEA [12], replacing Hf with Mo (TiZrNbTaMo) and reported lower corrosion potential compared to conventional alloys. Furthermore, Navi et al. [52] analyzed the polarization resistance and current density, indicating better performance of the high-entropy alloy. When comparing with the SS304, Shittu et al. [13] confirms better corrosion resistance for the high-entropy alloy, both for corrosion potential and current density. Akmal et al. [41], with a slightly different composition $(MoTa)_{0.2}NbTiZr$, also point to better performance of bioHEA when compared to cp-Ti and 316L. Analyzing the same alloy (MoNbTaTiZr), Perumal et al. [22] found that samples submitted to mechanical processing of FSP or SFP obtained a considerable increase in their resistance to polarization, in which the performance of HEA-SFP can be highlighted, with polarization resistance of 2207 $k\Omega cm^2$.

Peightambardoust et al. [59] evaluated Ti6Al4V, CoCrMo, and 316L substrates with 1025 µm coating of HEA $Ti_{1.5}ZrTa_{0.5}Nb_{0.5}Hf_{0.5}$, indicating success in improving anticorrosive properties.

In the study of HEA $Al_xCoCrFeNi$ (x = 0.6, 0.8 and 1) [62], different potentials for impedance spectroscopy (EIS) were used in the analysis of corrosion performance in saline environment infectious, with pH = 3 (Figure 9). In the content of 0.6 Al, the highest corrosion resistance was obtained due to the formation of a protective oxide layer, which was indicated by an increase in the low frequency impedance. Through the potential of +0.1 V, it was found that the alloy with 0.8 Al presented polarization resistance of about 6 $M\Omega cm^2$, the highest among HEAs. After increasing the potential to +0.7 V, $Al_{0.6}CoCrFeNi$ stood out with greater resistance: approximately 3.3 $M\Omega cm^2$ [62].

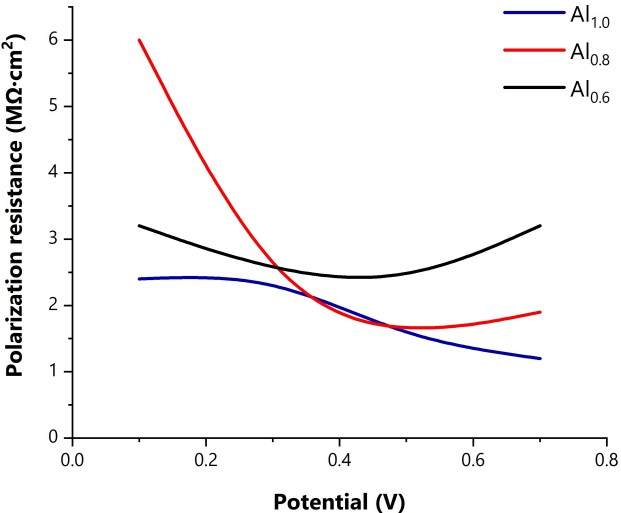

**Figure 9.** Comparison of polarization resistance of HEA $Al_xCoCrFeNi$ with different EIS potentials. Based on [62].

The surface corrosive behavior of HEA $(TiZrNbTa)_{90}Mo_{10}$ [69] was evaluated in Ringer's solution, observing that in all cases (pH 1, 3, 7, and 9), a protective oxide layer was formed. For pH = 1, the corrosion current density $i_{corr}$ and the polarization resistance $R_p$ showed that the passive film protection deteriorated considerably. With pH = 3, the highest polarization resistance value was obtained: 4.7 $M\Omega cm^2$ (Figure 10) [69].

**Table 8.** Anticorrosive performance of bioHEAs and conventional alloys.

| Alloy | Solution | $E_{corr}$ (mV) | $R_p$ (kΩcm²) | $I_{corr}$ (µA/cm²) | Reference |
|---|---|---|---|---|---|
| Ti6Al4V | | −526 | 261 | 0.18 | |
| 316L | | −216 | 52 | 1.32 | |
| CoCrMo | PBS | −331 | 163 | 0.28 | [12] |
| TiZrTaHfNb | | −391 | 554 | 0.07 | |
| $Ti_{1.5}ZrTa_{0.5}Hf_{0.5}Nb_{0.5}$ | | −396 | 780 | 0.06 | |
| Ti6Al4V | | −571 | - | - | |
| 316L | PBS | −234 | - | - | [50] |
| CoCrMo | | −320 | - | - | |
| TiZrNbTaMo | | −607 | - | - | |
| $Ti_{1.5}ZrTa_{0.5}Nb_{0.5}Hf_{0.5}$-Ti6Al4V | | −168 | 830 | 0.04 | |
| HEA-316L | PBS | −100 | 782 | 0.04 | [59] |
| HEA-CoCrMo | | −88 | 818 | 0.04 | |
| cp-Ti | | −420 | - | - | |
| 316L | PBS | −260 | - | - | [41] |
| $(MoTa)_{0.2}NbTiZr$ | | −530 | - | - | |
| Ti6Al4V | | −462 | 237 | 0.16 | |
| $Ti_{1.5}ZrTa_{0.5}Nb_{0.5}W_{0.5}$ | PBS | −160 | 306 | 0.09 | [58] |
| HEA-Ag | | −190 | 492 | 0.07 | |
| IM-CoCrFeCuNi | NaCl | - | 38 | - | [9] |
| SLM-CoCrFeCuNi | | - | 15 | - | |
| 316L Substrate | Ringer's solution | −79 | 600 | 0.03 | [66] |
| TiNbMoMnFe Film | | −59 | 900 | 0.02 | |
| CoNiCr | | −430 | - | 0.06 | |
| FeCoNiCr | Ringer's solution | −150 | - | 0.04 | [55] |
| FeCoNiCrPd | | 60 | - | 0.02 | |
| cp-Ti | | −250 | - | 0.26 | |
| TiZrNbHfSi | Ringer's solution | −330 | - | 0.08 | [24] |
| $Ti_{30}Zr_{25}Nb_{25}Si_{15}Ga_3B_2$ | | −250 | - | 0.15 | |
| $Ti_{20}Zr_{20}Nb_{20}Hf_{20}Si_{15}Ga_3B_2$ | | −320 | - | 0.13 | |
| CoCrMo | | −447 | - | 7.34 | |
| TiMoVWCr | Ringer's solution | −500 | - | 4.51 | [54] |
| TiMoVNbZr | | −481 | - | 2.13 | |
| $Al_{1.0}CrFeCoNi$ | | - | 1300–2200 [1] | | |
| $Al_{0.8}CrFeCoNi$ | Ringer's solution | - | 1900–6100 [1] | - | [62] |
| $Al_{0.6}CrFeCoNi$ | | - | 3200–3200 [1] | - | |
| $(TiZrNbTa)_{90}Mo_{10}$ | Ringer's solution | - | 850–4600 [2] | 0.01–0.07 [2] | [69] |
| Ti6Al4V | | −1150 | - | 3.89 | |
| $(TiZrNb)_{14}SnMo$ | SBF | −850 | - | 2.01 | [64] |
| HEA coating | | −1000 | - | 1.10 | |
| Ti6Al4V | SBF | −140 | 27 | 4.64 | [52] |
| TiNbTaZrMo | | −420 | 226 | 0.34 | |
| SS304 | SBF | −123 | - | 1.70 | [13] |
| MoNbTaTiZr | | −118 | - | 0.30 | |
| Ti6Al4V | Hank's solution | −325 | 677 | - | [45] |
| TiZrHfNbTa | | −395 | 642 | - | |
| MoNbTaTiZr | | −314 | 29 | 0.22 | |
| MoNbTaTiZr (FSP) | Ringer's solution and SBF | −175 | 450 | 0.11 | [22] |
| MoNbTaTiZr (SFP) | | −142 | 2207 | 0.04 | |
| AlCoCrFeNi | | −256 | - | 0.43 | |
| $AlCoCrFeNi_{1.4}$ | - | −200 | - | 0.75 | [63] |
| $AlCoCrFeNi_{1.8}$ | | −207 | - | 0.37 | |

[1] For different potential values (V). [2] For different pH values in the solution.

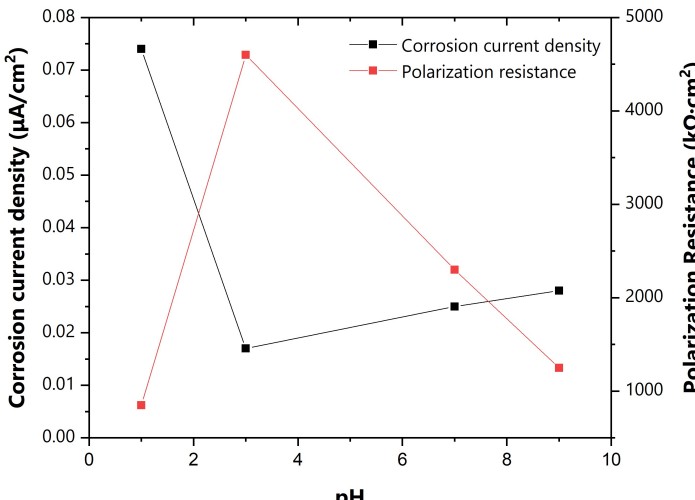

**Figure 10.** Corrosion current density and polarization resistance of HEA TiZrNbTaMo after 24 h exposure in Ringer's solution. Based on [69].

By analyzing the corrosive behavior of three alloys (TiTaHfNb, TiTaHfNbZr, and TiTaHfMoZr) [25] with immersion tests in fetal bovine serum (FBS) after 1, 7, 14, and 28 days, it was found that the highest release of the concentration of ions occurred in HEA TiTaHfNbZr, with approximately 400 ppb. Furthermore, the alloy containing Mo showed an increase in ion release after 28 days: about 360 ppb. Therefore, the addition of the elements Zr and Mo to the TiTaHf base caused the ion concentration to increase in FBS. On the other hand, TiTaHfNb showed a reduction in the release of ions and a concentration of 310 ppb after the immersion period, indicating a greater resistance to corrosion in FBS. Among the constituent elements, Ti showed the highest release of ions in the three alloys, with values of 309 ppb (TiTaHfNb), 347 ppb (TiTaHfNbZr), and 185 ppb (TiTaHfMoZr). The higher release of Ti may indicate the formation of a protective oxide layer in the samples, which was evidenced by the analysis of XPS [25].

With the three HEAs TiTaHfNb, TiTaHfNbZr, and TiTaHfMoZr [46], it was possible to observe that the presence of Zr and Nb improved the corrosion resistance performance of the samples. Through SEM micrographs after immersion in SBF and AS (artificial saliva), it appears that there was no significant corrosion on the surface of TiTaHfNbZr. On the other hand, the alloy without Zr (TiTaHfNb) showed corrosive behavior in SBF. For HEA with Mo, there was a large amount of the release of ions of this element. The analysis of the concentration of ions after immersion for 1, 7, 14, 21, and 28 days, demonstrate values much higher of release for the alloy with Mo, while the alloys with Nb and Zr proved to be more stable and with lower release of ions [46].

The good anticorrosive performance of high-entropy alloys compared to conventional alloys may also be related to the formation of a protective oxide layer, as reported in several works. Some examples are the HEAs Al$_x$CoCrFeNi (x = 0.6, 0.8 and 1) [62], TiZrTaHfNb [45], MoNbTaTiZr [22], (TiZrNb)$_{14}$SnMo [64], TiTaHfNb, TiTaHfNbZr and TiTaHfMoZr [46].

*3.2. Cell Viability*

Six works developed a study about cell density and viability comparing high-entropy alloys and conventional alloys. Overall, HEAs performed similarly [10,15,45,70] or better [22,53] to Ti alloys, and better than 316L and CoCrMo. A possible factor for obtaining a favorable microenvironment for cell adhesion and good density results may be the presence of Ti and Zr in the alloys evaluated in this section.

In assessing cell density, Iijima et al. [70], Ishimoto et al. [15], and Todai et al. [53] compared bioHEAs with alloys commonly used for biomedical applications, pointing out that high-entropy alloys have good performances, similar or superior to cp-Ti. Iijima et al. [70] reported

that $Ti_{28.33}Zr_{28.33}Hf_{28.33}Nb_{6.74}Ta_{6.74}Mo_{1.55}$ has a cell density of more than 7000 cells/cm². In the work by Ishimoto et al. [15], the SLM-HEA $Ti_{1.4}Nb_{0.6}Ta_{0.6}Zr_{1.4}Mo_{0.6}$ showed a density of more than 8000 cells/cm². This value is higher than the results obtained for the cast alloy of the same composition of about 7500 cells/cm². Finally, the publication by Todai et al. [53] pointed out that heat treatment improved the cell density results for the TiNbTaZrMo alloy from 100 to approximately 150 cells/mm².

Other papers have reported on the percentage of cell viability for bioHEAs. Yang et al. [45] showed the similarity between HEA TiZrHfNbTa and Ti6AL4V for cell viability during cell culture with different days of incubation. The result after 7 days was about 100% for both alloys. Perumal et al. [22] points out that samples of HEA MoNbTaTiZr processed by SFP and FSP performed better than the cast alloy, all with more than 90% of viable cells after incubation for 48 h. The values obtained for the high-entropy alloys also stand out against the Ti6Al4V and 316L alloys.

### 3.3. Antimicrobial Activity

One of the main causes of implant failure is bacterial infection [71], which makes a thorough evaluation of this aspect necessary for biomaterials. However, only one work was found that conducted a study on antimicrobial activity, becoming a field of opportunity for publications involving high-entropy alloys for biomedical applications.

In the comparison between the almost equiatomic HEA CoCrFeCuNi manufactured by selective laser melting (SLM) and the same alloy made by the traditional metallurgy process [9], the alloy by SLM obtained better antibacterial performance, as shown in Table 9. For antibacterial rates against *Escherichia coli* (*E. coli*), SLM-HEA showed 98% for both sessile and planktonic cells. The fused alloy performed 94% in sessile cells and 92% in planktonic cells [9].

**Table 9.** Comparison of antibacterial rates against *E. coli* by fused CoCrFeCuNi and by SLM.

| Alloy | Antibacterial Rates against *E. coli* | |
|---|---|---|
| | **In Sessile Cells** | **In Planktonic Cells** |
| IM-HEA | 94% | 92% |
| SLM-HEA | 98% | 98% |

Regarding the bacterium *Staphylococcus aureus* (*S. aureus*), after 24 h of inoculation more than 99% of the bacteria was eliminated in the alloy samples. The authors highlighted microbe-influenced corrosion as a major problem for biomaterials and with a high cost of damage [9]. Thus, adding elements such as Cu and Ag favors antibacterial performance. Analysis of the release of Cu ions containing *S. aureus* allowed an evaluation of the antibacterial efficacy of HEA, obtaining about 12 mg/L of release of Cu ions in the molten sample and 25 mg/L for the SLM sample. These results demonstrate a greater efficacy for the alloy obtained by selective laser melting. For the evaluation with E. coli, the SLM sample also excelled [9].

### 3.4. Magnetic Susceptibility

The HEA MoNbTaTiZr [18] was evaluated for use in implants that will be subjected to magnetic resonance, with an analysis of the magnetic susceptibility of the HEA that presented a value of approximately $2 \times 10^{-6}$ dM/dH(cm³/g). The authors highlight that the Zr element is responsible for decreasing the value of this property.

Calin et al. [24] compared magnetic susceptibility volume for bioHEAs and conventional alloys, highlighting better performance for TiZrNbHfSi and TiZrNbSiGaB alloys (Table 10). The values obtained make it possible to manufacture and apply biomaterials with these compositions, as they do not significantly hinder medical follow-up with magnetic resonance imaging, such as stainless steel, for example [24].

**Table 10.** Corrosion current density and magnetic susceptibility volume for TiZrNbHfSi, non-equiatomic TiZrNbSiGaB, and non-equiatomic TiZrNbHfSiGaB HEAs.

| Alloy | Magnetic Susceptibility Volume ($X_v$) [ppm] |
|---|---|
| cp-Ti | 182 |
| 316L | 3520–6700 |
| TiZrNbHfSi | 50 |
| TiZrNbSiGaB | 46 |
| TiZrNbHfSiGaB | 191 |

## 4. Conclusions

This review presents an assessment of the use of HEAs in biological applications. Based on what is presented, bioHEAs have advantages over conventional biomedical alloys, and could complement Ti6Al4V, CoCrMo, cp-Ti, and 316L in key situations. From this study, the conclusions are as follows:

- Among the bioHEAs structures, the simple BCC was the most obtained, based on elements such as Ti, Ta, and Hf;
- Although composition is relevant to material hardness, the processing and heat treatment of bioHEAs proved to be more influential for this property. Amorphous HEAs that were used as a coating on conventional alloy substrates showed high hardness [58]. On the other hand, the most suitable Young's modulus for biomedical applications was found in BCC structures [19,48];
- It is noteworthy that Ti, Nb, and Ta hinder the corrosion dissolution [66,69], allowing greater resistance. In conjunction with this factor, the formation of a protective oxide layer helped in the performance against corrosion [12,45,62,64]. For antibacterial characteristics, Ag, Cu, and Zn are promising elements that can bring good results [9,67];
- Compared to conventional biomedical alloys, high-entropy alloys presented interesting mechanical, chemical, and biological properties in most cases evaluated in this review;
- In the biocompatibility analyses, the predominance of corrosive tests of the alloy was verified. Antibacterial performance, viability, cell density and adhesion, and magnetic susceptibility are other assays performed for bioHEAs;
- Because the study of high-entropy alloys for biomedical applications is a relatively new and current topic, there is a need to evaluate the biological influence of these alloys in long-term applications.

**Author Contributions:** Conceptualization, T.G.d.O. and A.A.A.P.d.S.; methodology, D.S. and A.A.A.P.d.S.; software, T.G.d.O.; validation, T.G.d.O. and A.A.A.P.d.S.; formal analysis, T.G.d.O.; investigation, T.G.d.O.; resources, D.S and A.A.A.P.d.S.; data curation, D.V.F. and A.A.A.P.d.S.; writing—original draft preparation, T.G.d.O.; writing—review and editing, D.V.F., P.C. and A.A.A.P.d.S.; visualization, T.G.d.O. and A.A.A.P.d.S.; supervision, P.C. and A.A.A.P.d.S.; project administration, T.G.d.O.; funding acquisition, A.A.A.P.d.S. All authors have read and agreed to the published version of the manuscript.

**Funding:** This research received no external funding.

**Data Availability Statement:** Not applicable.

**Conflicts of Interest:** The authors declare no conflict of interest.

## Abbreviations

The following abbreviations are used in this manuscript:

| | |
|---|---|
| HEA | High-entropy alloy |
| bioHEA | bio-high-entropy alloy |
| SLM | Selective laser melting |
| BCC | Body-centered cubic |
| HPT | High pressure torsion |
| HT | Heat treatment |
| VAM | Vacuum arc melting |
| VAR | Vacuum arc remelting |
| SPS | Spark plasma sintering |
| DFT | Density functional theory |
| SEM | Scanning electron microscopy |
| SFP | Stationary friction processing |
| FCC | Face centered cubic |
| LPS | Liquid phase separation |
| HCP | Hexagonal close-packed |
| CP | Primitive cubic |
| PBS | Phosphate buffer solution |
| FBS | Fetal bovine serum |
| SBF | Simulated body fluid |
| AS | Artificial saliva |

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
