# Peer review of "A Review of Biomaterials Based on High-Entropy Alloys"

_metals, doi:10.3390/met12111940_

Round 1
Reviewer 1 Report
This paper reports a study on HEAs and specifically their application in biomedical field. The subject is interesting and the report contains useful and helpful information that can be used to understand the advances of HEAs in biomedical field. The manuscript is well roganized and the main subjects are well connected. However, I could notice that the manuscript contains many grammer and expression mistakes that should be addressed before pubishing. For example:
1- First line of abstract says "Due to it great..." should be Due to its great...
2- Line 37 of page 2 "present artigle" should be article.
3- Expression mistake in line 40 of page 2 " areas of its application biomedicine" should be biomedical.
and many more that someone can notice. Therefore, I would recommend publication after minor revision and review by native english speaker.
Author Response
Dear Editor and Reviewer,
The authors of the manuscript and I are very grateful for your review and considerations. Rest assured that all of your observations were carefully analyzed and they undoubtedly enriched the result of the work. Thus, you can find comments on the revisions made based on your review following.
This paper reports a study on HEAs and specifically their application in biomedical field. The subject is interesting and the report contains useful and helpful information that can be used to understand the advances of HEAs in biomedical field. The manuscript is well organized and the main subjects are well connected. However, I could notice that the manuscript contains many grammar and expression mistakes that should be addressed before publishing. For example:
1- First line of abstract says "Due to it great..." should be Due to its great...
– The text was modified accordingly.
2- Line 37 of page 2 "present artigle" should be article.
– The text was modified accordingly.
3- Expression mistake in line 40 of page 2 " areas of its application biomedicine" should be biomedical.
– The text was modified accordingly.
and many more that someone can notice. Therefore, I would recommend publication after minor revision and review by native english speaker.
– We thank the reviewer for the comments and considering our work as worthwhile for publication in Metals. Besides, the manuscript was reviewed and others typing errors were ghostly corrected in the manuscript.
Again, I appreciate your availability for reviewing and indicating improvements in the work. If you have any questions or other suggestions, please do not hesitate to contact me.
Reviewer 2 Report
The manuscript entitled A review of biomaterials based on high-entropy alloys written by Thiago Oliveira et al. was presented for review. It has a review character. It should be emphasized that the subject of the paper is interesting and relevant from both a practical and scientific point of view. Therefore, literature related to high-entropy alloys is relatively rich in review-type articles. For this reason, prior to subsequent reviews, an original approach to the issue is required, so that the reader is interested in and provided with current knowledge of the subject. The Authors do not delve too deeply into the presented problems but briefly present the main ideas worth mentioning. This approach can be questionable. Moreover, I found no reason why this review is better than the others. In my opinion, the paper is not original and interesting enough to be published in Metals.
Author Response
25th October 2022
Dear Editor and Reviewer,
The authors of the manuscript and I are very grateful for your review and considerations. Rest assured that all of your observations were carefully analyzed and they undoubtedly enriched the result of the work. Thus, you can find comments on the revisions made based on your review following.
The manuscript entitled A review of biomaterials based on high-entropy alloys written by Thiago Oliveira et al. was presented for review. It has a review character. It should be emphasized that the subject of the paper is interesting and relevant from both a practical and scientific point of view. Therefore, literature related to high-entropy alloys is relatively rich in review-type articles. For this reason, prior to subsequent reviews, an original approach to the issue is required, so that the reader is interested in and provided with current knowledge of the subject. The Authors do not delve too deeply into the presented problems but briefly present the main ideas worth mentioning. This approach can be questionable. Moreover, I found no reason why this review is better than the others. In my opinion, the paper is not original and interesting enough to be published in Metals.
– We thank the reviewer for reading our work and addressing the comments. In fact, there are several review-type articles available considering HEA in the literature. However, as far as we know, only two considering its potential application as biomaterials (listed below). Ahmady et al. (2023) presents a review focused in HEAs applied as coatings, while Castro et al. (2021) an overview of potential applications mechanical and a brief section of biocompatibility studies. In our view, there is an important lack in the literature concerning the microstructural, biological and chemical properties approach that we are trying to fill up. Hoping to make it clearer, we have included a new paragraph in the end of section 1 (p.3 l.72-77.).
D Castro, P Jaeger, AC Baptista, JP Oliveira. An Overview of High-Entropy Alloys as Biomaterials, Metals, 2021. https://doi.org/10.3390/met1104064
AR Ahmady, A Ekhlasi, A Nouri, MH Nazarpak, P Gong, A Solouk, High entropy alloy coatings for biomedical applications: A review. Smart Materials in Manufacturing, 2023. https://doi.org/10.1016/j.smmf.2022.100009
Again, I appreciate your availability for reviewing and indicating improvements in the work. If you have any questions or other suggestions, please do not hesitate to contact me.
Sincerely,
Patricia
Reviewer 3 Report
1) The text should be modifying to english ; Another , both "Abstract:" and " Conclusions" isnt focuse on bio-HEAs.
2) The text is checking, lazy to explain, ex., Eq. 1: l & n, style problem, cp-Ti & Cp-Ti, Arc-melting & Arc melting, BioHEAs & BioHEA, double CFC & double FCC..
3) " most of the experimental works used the 62 arc or induction melting techniques (Figure 1),..", However, 73% (AR),11% (MA), 8% (IN) is main tech. another, Why is the analysis of %?
Author Response
25th October 2022
Dear Editor and Reviewer,
The authors of the manuscript and I are very grateful for your review and considerations. Rest assured that all of your observations were carefully analyzed and they undoubtedly enriched the result of the work. Thus, you can find comments on the revisions made based on your review following.
1) The text should be modifying to english ; Another , both "Abstract:" and " Conclusions" isnt focuse on bio-HEAs.
– The manuscript was reviewed, the English grammar was improved, and others typing errors were ghostly corrected in the manuscript. Both abstract and conclusions sections were modified to address the reviewer’s comments.
2) The text is checking, lazy to explain, ex., Eq. 1: l & n, style problem, cp-Ti & Cp-Ti, Arc-melting & Arc melting, BioHEAs & BioHEA, double CFC & double FCC..
– The text was modified accordingly.
3) " most of the experimental works used the 62 arc or induction melting techniques (Figure 1),..", However, 73% (AR),11% (MA), 8% (IN) is main tech. another, Why is the analysis of %?
– The reviewer is right, and text was modified to make this statement clearer (pg. 2 l. 64-65).
“Regarding the development of bioHEAs, most of the experimental works used the melting techniques…”
Again, I appreciate your availability for reviewing and indicating improvements in the work. If you have any questions or other suggestions, please do not hesitate to contact me.
Sincerely,
Patricia
Reviewer 4 Report
In short, the authors need to drastically narrow the scope of this review—A thorough description of conventional biocompatible alloys should be given with respect to their mechanical properties, fatigue performance, corrosion resistance, etc. The authors should then specifically discuss in detail how one-to-one comparisons can be made between the existing biomaterials in production and the MPEAs presented in this study. Simply dropping everything into tables is not enough.
The authors state “at the end of the 1990s a new class of material was developed” however fail to cite any publications relating to the 1990s.
The authors should also describe the usage of the “multi-principal element alloy” (MPEA) nomenclature.
Limiting the percentage from 5-35% is entirely arbitrary and not rooted in science.
HEAs “can” be defined by configurational entropy but this also does nothing to actually describe the complex metallic bonding that can take place.
The “four core” effects are not very well described by Ref. 5, and on that note, why is this review being written as it seems to duplicate the efforts of Ref. 5?
Line 39: Due to “its” outstanding properties and thermodynamic stability…” – this is not a general statement, and misleading to the reader. The overwhelming majority of HEA/MPEA combinations are not viable for biomedical usage. If the authors are discussing one alloy in particular, state as such.
On that note, references 6 – 21 in that paragraph are meaningless as HEAs are not a “class” of materials per se—this is a blanket term used to describe alloys that predominantly have no base element, where the common attributes tend to be substitutional solid solutions in one or several phases. Reading this paragraph within the context of “alloys” as opposed to “high-entropy alloys” makes it lack context, and is therefore useless unless the authors describe which alloy systems in particular these references allude to. And not only that, but why the comparison is even there at all? This would be as meaningless as an article that discusses the benefits of “alloys” and compare how great they are by including references to stainless steels, carbides, aluminum alloys, titanium alloys etc… yes, we know alloys are useful, however, explain to the reader with more context or else this information is not useful.
Table 1: why was young’s modulus and hardness chosen to compare these alloys for biomedical applications? Usually when Ti-6Al-4V is discussed with respect to biomedical applications, fatigue strength/performance is used.
Table 2 is meaningless. Why is Ni even included at all? Nickel and nickel compounds are considered carcinogenic to humans. NiTi alloys have to undergoe rigorous testing to ensure a limitation of Ni-ion release due to corrosion.
That being said, the authors should list a table of each of these elements and their corresponding carcinogenic potential and cite the source used. The authors should also include a table of common biocompatible alloys that are already in production.
Line 56: The authors use the term “bioHEA” but provide not definition. What is a bioHEA? An HEA that others think can be biocompatible? Who is the authority on this definition?
Table 3: Why is single phase important? One of the most critically used alloys in biomaterial applications is Ti-6Al-4V, which is a 2-phase alpha/beta alloy. Why is BCC important? The authors place too much importance on “single phase” HEAs when most alloys that are used in industry are not single phase.
The authors use the term “SLM” without providing a definition to what this is. “Selective laser melting” I assume.
With Tables 3 and 4, the author should also describe the post-processing (if any)
Figure 5. Please do not connect the dots. Draw trendlines. Also, how many datapoints were collected for these values. Were they averaged? Were the datasets analyzed for statistical significance using analysis of variance techniques? That is, is there a statistical difference between the datasets?
Table 5. Ni is in all of these. See my previous comment about Ni carcinogenicity and the testing that NiTi (nitinol) alloys undergo.
It should be noted that arc-melted is not an equilibrium process, and the microstructures presented may not be representative of the equilibrium solidification phases.
Author Response
25th October 2022
Dear Editor and Reviewer,
The authors of the manuscript and I are very grateful for your review and considerations. Rest assured that all of your observations were carefully analyzed and they undoubtedly enriched the result of the work. Thus, you can find comments on the revisions made based on your review following.
In short, the authors need to drastically narrow the scope of this review—A thorough description of conventional biocompatible alloys should be given with respect to their mechanical properties, fatigue performance, corrosion resistance, etc. The authors should then specifically discuss in detail how one-to-one comparisons can be made between the existing biomaterials in production and the MPEAs presented in this study. Simply dropping everything into tables is not enough.
– We have included two columns in table 1 to address that, one with the YS and other with TS. However, it should be pointed out, that mechanical characterization of alloys classified as HEAs are barely available in the literature and in the rare cases that they accessible we tried to compare (e.g., p.7, l.175-176; p.9, l.238-239; p.9, l.245-246). Besides, Hardness and Young modulus are considered more important for biological applications once YS of metallic materials, normally is far beyond higher them the necessary for this application.
Concerning the corrosion resistance, we consider that making a parallel for all alloys presented one by one would be tedious and meaningless for the reader. Based on that we compared the values obtained for the HEAs along with the classical materials when the results were remarkable, e.g.:
(1) p.16, l.356-359 (“Motallebzadeh et al. [13] evaluated two TiZrTaHfNb-based alloys with different compositions. These alloys presented a lower performance compared to Ti6Al4V, but with better polarization resistance.”);
(2) p.16, l.365-370 (“Furthermore, Navi et al. [51] analyzed the polarization resistance and current density, indicating better performance of the high-entropy alloy. When comparing with the SS304, Shittu et al. [14] confirms better corrosion resistance for the high-entropy alloy, both for corrosion potential and current density. Akmal et al. [41] with slightly different composition (MoTa)0.2NbTiZr also points to better performance of bioHEA compared to cp-Ti and 316L.”);
(3) p.16, l.374-376 (“Peightambardoust et al. [58] evaluated Ti6Al4V, CoCrMo and 316L substrates with 1025 µm coating of HEA Ti1.5ZrTa0.5Nb0.5Hf0.5, indicating success in improving anticorrosive properties.”);
(4) p.18 l.411-415 (“The good anticorrosive performance of high-entropy alloys compared to conventional alloys may also be related to the formation of a protective oxide layer, as reported in several works. Some examples are the HEAs AlxCoCrFeNi (x = 0.6, 0.8 and 1) [61], TiZrTaHfNb [45], MoNbTaTiZr [23], (TiZrNb)14SnMo [63], TiTaHfNb, TiTaHfNbZr and TiTaHfMoZr [46].”)
The authors state “at the end of the 1990s a new class of material was developed” however fail to cite any publications relating to the 1990s.
– Taking into account the reviewer’s comments, we have included the articles from Yeh et al. (2004) and Cantor et al. (2004), that are stated as the pioneer’s published articles concerning HEAs.
The authors should also describe the usage of the “multi-principal element alloy” (MPEA) nomenclature.
– In order to take into account, the reviewer’s comments, we have subtle modified the first paragraph of the manuscript’s introduction. (p.1, l.17-24)
“In the early 2000’s a new class of material was developed and aroused interest in the research community [1,2], leading to an exponential number of publications in the last two decades, they are widely known as high-entropy alloys (HEAs) or multi-principal element alloy (MPEA). Although, precise definition of a HEA still controversial, it seems to be consensus among the authors that these alloys should be composed by at least four main elements with concentration between 5 and 35 at% [2–7], in contrast to the conventional alloys that are based on a main element (e.g., Fe for steel, Ni or Co in superalloys, Cu for bronze and brass etc)”.
Limiting the percentage from 5-35% is entirely arbitrary and not rooted in science.
– The reviewer has a point. However, this definition is widely accepted in the community. JW Yeh first state this concept, in his US patent application No US2002159914-A1 [22]: “The features of the alloys are that there are five to eleven major metallic elements and with or without minor elements, the minor elements are selected from the element group other than the major metallic elements, the mole fraction of each major metallic element in the alloy is between 5% and 30%, and the mole fraction of each minor element in the alloy is less than 3.5%.”. Later, Senkov et al. (2018) distinguish between ‘refractory high entropy alloys’ (RHEAs) “consisting of five or more principal elements with concentrations between 5% and 35%, and ‘refractory complex concentrated alloys’ (RCCAs), containing three or more elements with concentrations> 35%.” Several other authors included similar definition in their papers. As we stated (pg.1 l.22) the definition of HEA or MEPA is still controversial due to it youngness, it is beyond the scope of this present article “untie this knot”. In order support our statement, we have included other author in the reference that concerns this sentence.
HEAs “can” be defined by configurational entropy but this also does nothing to actually describe the complex metallic bonding that can take place.
– We are sorry, but we were not able to understand the reviewer’s question.
The “four core” effects are not very well described by Ref. 5, and on that note, why is this review being written as it seems to duplicate the efforts of Ref. 5?
– The reviewer is right. We have substituted the reference to another that we find more appropriated. Concerning contrast of the contribution of the present work and the Castro’s et al, Castro’s work is an overview of potential applications mechanical and a brief section of biocompatibility studies. In our view, there is an important lack in the literature concerning the biological and chemical properties and microstructural approach that we are trying to fill up. Hoping to make it clearer, we have included a new paragraph in the end of section 1 (p.3 l.74-79.).
Line 39: Due to “its” outstanding properties and thermodynamic stability…” – this is not a general statement, and misleading to the reader. The overwhelming majority of HEA/MPEA combinations are not viable for biomedical usage. If the authors are discussing one alloy in particular, state as such.
– We agree with the reviewer’s comment. The sentence was imprecise, and we have modified in order to address this issue (p.2 l.38).
On that note, references 6 – 21 in that paragraph are meaningless as HEAs are not a “class” of materials per se—this is a blanket term used to describe alloys that predominantly have no base element, where the common attributes tend to be substitutional solid solutions in one or several phases. Reading this paragraph within the context of “alloys” as opposed to “high-entropy alloys” makes it lack context, and is therefore useless unless the authors describe which alloy systems in particular these references allude to. And not only that, but why the comparison is even there at all? This would be as meaningless as an article that discusses the benefits of “alloys” and compare how great they are by including references to stainless steels, carbides, aluminum alloys, titanium alloys etc… yes, we know alloys are useful, however, explain to the reader with more context or else this information is not useful.
– In order to take into account, the reviewer’s comments, we have subtle modified the following paragraph of the manuscript’s introduction. (p.2, l.38-45)
Table 1: why was young’s modulus and hardness chosen to compare these alloys for biomedical applications? Usually when Ti-6Al-4V is discussed with respect to biomedical applications, fatigue strength/performance is used.
– We have included two columns in table 1 to address that, one with the YS and other with TS. However, it should be pointed out, that mechanical characterization of alloys classified as HEAs are barely available in the literature and in the rare cases that they accessible we tried to compare (e.g., p.7, l.175-176; p.9, l.238-239; p.9, l.245-246). Besides, Hardness and Young modulus are considered more important for biological applications once YS of metallic materials, normally is far beyond higher them the necessary for this application.
Table 2 is meaningless. Why is Ni even included at all? Nickel and nickel compounds are considered carcinogenic to humans. NiTi alloys have to undergoe rigorous testing to ensure a limitation of Ni-ion release due to corrosion.
That being said, the authors should list a table of each of these elements and their corresponding carcinogenic potential and cite the source used. The authors should also include a table of common biocompatible alloys that are already in production.
– We disagree that the table 2 is meaningless. With the large number of alloys classified of HEA and assessed to be used in biomedical applications, the purpose of the table is to guide the reader into the most investigated compositions.
Concerning the presence of Ni in these alloys, the reviewer has a point, and we have included Ni in the list of barely used elements (p.2 l. 61) and highlighted that in the text. However, we should remember that the aim of the present work is to give the community a review of the investigations’ guidelines concerning HEA and its Biomedical applications.
Line 56: The authors use the term “bioHEA” but provide not definition. What is a bioHEA? An HEA that others think can be biocompatible? Who is the authority on this definition?
– In order to clarify the reviewer’s comments, we have included the following sentence in the manuscript (p.2 l.38-40):
“In 2019, the same group published two articles [10,11] using the term bioHEA as far as known for the first time. Since them, this term is applied for the multi-principal element alloys that have been considered for application in biomedical area.”
Table 3: Why is single phase important? One of the most critically used alloys in biomaterial applications is Ti-6Al-4V, which is a 2-phase alpha/beta alloy. Why is BCC important? The authors place too much importance on “single phase” HEAs when most alloys that are used in industry are not single phase.
– Due to its high entropy, this class of alloys trends to have simple solid solution type microstructure where BCC is the most common. To present the key information available in the literature, as organized as possible to the reader, we have grouped the alloys by microstructural features, i.e., the phases present in the alloys, starting to the most common, the BCC single phase alloy.
The authors use the term “SLM” without providing a definition to what this is. “Selective laser melting” I assume.
– The term “selective laser melting” is defined in p.2 l.68-69 and in the “abbreviations” section (p. 20)
With Tables 3 and 4, the author should also describe the post-processing (if any)
– We have added a column in the tables including that information
Figure 5. Please do not connect the dots. Draw trendlines. Also, how many datapoints were collected for these values. Were they averaged? Were the datasets analyzed for statistical significance using analysis of variance techniques? That is, is there a statistical difference between the datasets?
– In order to take into account, the reviewer’s comments, we have modified Figure 5. Concerning the important questions of the reviewer, the original paper does not inform how many measurements were taken from the hardness tests and the young modulus and YS were measured based on compression tests performed in just one sample. We have included one sentence in other to clarify that (p.7, l.202-204).
Table 5. Ni is in all of these. See my previous comment about Ni carcinogenicity and the testing that NiTi (nitinol) alloys undergo.
– Concerning the presence of Ni in these alloys, the reviewer has a point, and we have included Ni in the list of barely used elements (p.2 l. 61) and highlighted that in the text. However, we should remember that the aim of the present work is to give the community a review of the investigations’ guidelines concerning HEA and its Biomedical applications.
It should be noted that arc-melted is not an equilibrium process, and the microstructures presented may not be representative of the equilibrium solidification phases.
– We agree with the reviewer and we are aware of that.
Again, I appreciate your availability for reviewing and indicating improvements in the work. If you have any questions or other suggestions, please do not hesitate to contact me.
Sincerely,
Patricia
Round 2
Reviewer 2 Report
I have decided to change my decision. It should be published in the present form.
Reviewer 4 Report
accept